environmental chemistry/hydrology/geophysics

remote sensing, bauxite residues, tailings moisture content, drying kinetics, reflectivity measurements, particulate emissions

**Author for correspondence:**
Josée Maurais
e-mail: Josee.Maurais@USherbrooke.ca

# Monitoring moisture content and evaporation kinetics from mine slurries through albedo measurements to help predict and prevent dust emissions

Josée Maurais, Frédéric Orban, Emrik Dauphinais and Patrick Ayotte

Département de chimie, Université de Sherbrooke, 2500 Boul. de l'Université, Sherbrooke (Qc), Canada J1 K 2R1

(iD) JM, 0000-0003-1575-9637

The prediction and prevention of fugitive dust emissions from mine tailings surfaces depend largely on our ability to monitor and monitor and predict the evolution of tailings moisture content (TMC). Albedo measurements are demonstrated here to be valuable tools to quantify TMC in bauxite residue samples under controlled conditions in the laboratory. The difference in albedo between 1.30 and 1.55 µm obtained through the infrared integrating sphere method shows good correlations with those acquired with a field spectroradiometer while both are strongly correlated with TMC. Additionally, continuous spectroscopic characterization of evaporating residues is shown to reveal the evolution in their surface drying rates. These optical methods could help predict surface drying state, thereby improving the accuracy of dust emissions risk assessment protocols that support mining industries intervention and mitigation strategies.

## 1. Introduction

The prevention and mitigation of fugitive dust emissions are environmental challenges that most mining industries face due to slurries and residues being stored in a vast open disposal area, as they are left outdoors to dry, exposed to the elements [1–3]. The dispersion of fugitive dust emissions emanating from tailings disposal sites over the surrounding communities and

neighbouring urban areas can become a concern as it may undermine mining industries' social acceptability, especially when emission standards and regulations are occasionally exceeded [4].

Several common, costly and labour-intensive methods are currently used to control and suppress fugitive dust emissions; namely, resorting to liquid dust suppressants or the application of sand or wood chips overlayers [5]. In addition to their costs and labour requirements, another key disadvantage of these methods is that the volume occupied by these materials decreases the mine tailings disposal areas storage capacity and thus, their life span [3,6–8]. Furthermore, as dust suppressants may inhibit bulk evaporation, systematically resorting to these physical methods is not desired in the long term. Indeed, slurries are required to dry until they reach solid fractions suitable for subsequent on-site geotechnical applications, therefore resorting to an excessive usage of dust suppressants may hamper evaporation and thus delay revalorisation and restoration of the disposal sites.

However, free evaporation from tailings can result in premature drying of the superficial layers which, in turn, favours the occurrence of fugitive dust emissions when capillary forces release their grip onto dust particles in the superficial layers [5,9,10]. Unfortunately, predicting the evolution of slurries' surface drying state is undermined by our poor understanding of the dependence of their evaporation kinetics on environmental conditions (temperature, relative humidity, illumination, precipitations, etc.) [10,11]. Therefore, improved continuous and surface sensitive and selective methods are required to monitor tailings moisture contents (TMC) for risk assessment purposes and thus support the mining industry in its efforts to predict and control or better yet, prevent and suppress fugitive dust emissions. Furthermore, they may also allow strategies to reduce liquid dust suppressants usage, and/or the application of physical overlayers (sand, wood chips), to be devised and optimized.

To tackle this problem, a bauxite residues storage area (BRSA) was selected as a case study to develop remote sensing monitoring tools using red mud samples and moisture contents that are representative of those found in the field, and under parameters relevant to actual disposal site operational and environmental conditions. Samples were collected from several sites at the BRSA, prepared for their physico-chemical characterization, including the quantification of their moisture content. Their surface drying states, and their evaporation kinetics, were scrutinized using surface specific optical techniques, namely, albedo measurements using a field spectroradiometer or the infrared integrating sphere (IRIS) method at short-wave infrared (SWIR) laser diode source wavelengths (i.e. 1.30 and 1.55 μm), as well as diffuse reflectance infrared Fourier transform spectroscopy (DRIFTS). These optical methods must be validated for their reliability and robustness for continuous monitoring of surface TMC under the harsh conditions of BRSA operations but also, provide sufficient precision, accuracy, surface sensitivity and selectivity for the intended remote sensing applications. Furthermore, they should ideally not rely on the highly variable solar irradiance flux as a light source to ensure robust and reliable TMC determinations all along the diurnal cycle. Previously reported methods to estimate soil moisture content include *in situ* measurements (gravimetric, dielectric measurements, etc.) [12–16], as well as remote sensing approaches, at radar [17–25], infrared [26,27] and optical [18,28–35] wavelengths that are known to probe water absorption bands. Unfortunately, most *in situ* methods either lack the sensitivity or selectivity towards the surface layer, or cannot be easily deployed for continuous real-time monitoring and field applications [13,36–38]. Therefore, most of them are unfortunately unsuitable for the specific context of evaluating mine slurries surface drying states and rates, *in situ* and in real time, as part of risk assessment protocols for dust emissions from tailings disposal sites.

Satellite-based remote sensing methods, on the other hand, possess several of these desirable attributes; however, they unfortunately suffer from atmospheric interferences due to absorption from water vapour, clouds and aerosols, which can severely compromise TMC measurements [31,39–41]. Another important limitation of satellite-based remote sensing methods is the influence of vegetation and canopy on optical soil moisture content measurements [30,39,42,43]. Furthermore, their spatial and temporal resolution as well as coverage are often insufficient to provide mine tailings disposal sites adequate resolving power and frequency (i.e. revisit period) to continuously monitor the evolution of surface TMC at the scale of individual plots [44–47]. Indeed, disposal sites can have areas of up to a few square kilometres and present dramatic changes in their surface drying state over just a few hours time, and a few meters distance, depending on specific local tailings physico-chemical properties, along with instantaneous meteorological conditions as well as their recent history [27].

At the BRSA selected for this study, bauxite residues are required to dry for up to four years and cover a 2 km$^2$ area, compartmentalized into more than 10 plots that display bulk solid fractions ranging between 50% and 75%. Individual tailing ponds present very heterogeneous properties and must thus

be monitored independently, necessitating measurement resolutions on the order of a few square meters, at intervals of tens of minutes [7]. Therefore, the methods described herein could supplement and complement remote sensing approaches, by providing validation data such as ground-based instantaneous TMC measurements at high time and space resolutions, necessary information to improve forecasting and mitigation of fugitive dust emission events from mining residues storage sites.

In this work, we show that the instantaneous surface TMC for bauxite residues sampled from the BRSA could be determined using optical methods. These latter may provide mining industries with more accurate and robust, surface sensitive and selective tools for the prediction and prevention of fugitive dust emissions from their tailings storage facilities as they allow for continuous monitoring of their surface drying state and rates using remote sensing techniques. Collectively, the approaches described herein should improve the ability of the mining industry to devise and implement efficient risk assessment protocols to assist in their residues management and environmental mitigation efforts.

# 2. Methods

## 2.1. Sample preparation

Raw bauxite residues were provided by the Centre de recherche et développement de l'aluminium (CRDA, Rio Tinto) and washed clean of caustic residues with water. A random sampling method was used at each site of the disposal area that was selected to be representative of the different drying stages. They were then oven dried at 200°C for 12 h and kept in airtight sealed containers until sample preparation and analysis.

Two different methods were used to prepare samples with selected bulk TMC (hereafter referred to as 'prepared samples') as the texture of humidified bauxite residues changes significantly above 33% $m_w/m_t$ moisture content ($m_w$: water mass; $m_t$: sample's total mass). Samples with TMC greater than 33% $m_w/m_t$ (i.e. solid fractions smaller than 67% $m_b/m_t$; $m_b$: dry bauxite tailing mass) form a viscous fluid. They were prepared by sequentially adding 1 to 2 g aliquots of water to an initially dry bauxite residue sample, followed by successive re-homogenising by mechanical stirring, prior to assessing their surface moisture content by way of the optical methods described below. Soon after mechanical stirring, samples prepared to have a moisture content greater than 50% $m_w/m_t$ settled rapidly due to sedimentation causing the samples' surfaces to display a noticeable supernatant layer. Samples with moisture contents smaller than 33% $m_w/m_t$ (i.e. solid fractions in excess of 67% $m_b/m_t$) retain their crumbly, granular texture and were prepared by stirring mechanically a known amount of bauxite residues (20, 25 or 30 g) with a known amount of water (1–15 g) until a macroscopically homogeneous mixture was obtained. Freshly prepared samples were transferred immediately to Petri dishes for analysis.

Using the sample preparation protocol described in the previous paragraph, samples intended for monitoring evaporation kinetics were prepared from a known amount of bauxite residues (5, 6, 15 or 30 g) and the required amount of water to reach 33 to 60% $m_w/m_t$ initial TMC (thereafter referred to as 'evaporating samples'). When sample evaporation approached completion, whereby the samples moisture content reached steady state conditions with ambient atmospheric humidity levels, the samples thickness ranged between about 1 and 4 mm. Wet bauxite residues evaporation was deemed complete when the water absorption feature in the spectral albedo (see below), as well as the total sample weight, reached their asymptotic values (i.e. did not evolve, within experimental precision, for more than 10 consecutive minutes). The evaporated sample mass was then acquired enabling the final TMC to be established.

## 2.2. Measurements

Spectral albedo measurements were performed using an ASD FieldSpec Pro spectroradiometer (Malvern PanAnalytical) equipped with a 1° aperture lens and calibrated using a spectralon with a reflectance of 99%. During measurements, the fibre detector sensing area was located at 20 cm from either the sample's or the spectralon's surface. Given this configuration and geometry, the field of view of the instrument is a disc of about 0.35 cm in diameter at the chosen working distance. The samples and spectralons were illuminated using a 600 W halogen lamp, located at 40 cm from their surface, and at an incidence angle of 15°. Each reported spectrum was acquired by averaging 20 consecutive spectra.

For prepared samples having selected TMC, four different positions on the surface of 30 samples were analysed and their spectral albedo averaged, that is the centre of the Petri dish along with three other, non-overlapping positions around the centre position. For evaporating samples, 20 consecutive spectral albedo measurements were acquired and averaged. This was repeated at 1 min intervals, and the surface TMC during sample evaporation was thus monitored continuously until it reached a steady state with the relative humidity of ambient air (42–67%) for eight different samples. The evaporation from the sample was deemed complete, either when the amplitude of the water absorption features in the spectral albedo data reached their asymptotic values, or when no measurable weight change could be detected, within experimental precision, for more than 10 consecutive minutes.

Surface TMC were also evaluated by single wavelength albedo measurements (i.e. at 1300 nm and 1550 nm), using the IRIS method, a photovoltaic detector and SWIR laser diodes [48]. Resorting to artificial light sources is preferred as it decreases the vulnerability of the measurements to natural fluctuations in solar irradiance and allows measurements throughout the diurnal cycle (as fugitive dust emissions are known to also occur at night). The samples and spectralons were placed, in alternation, under the integrating sphere, with either of the two different laser diode sources positioned at the nadir, and the detector located on the side (i.e. equatorial position) of the integrating sphere. Calibration of the IRIS apparatus was performed, separately for both laser diode sources, using five Lambertian spectralons (i.e. with 5, 20, 50, 75 and 99% reflectance), and was proven to be very reproducible and consistent over periods of several days as well as upon changing the diode laser sources. The sides of the Petri dishes used to hold the wet bauxite residues samples were covered with black electric tape to prevent stray light reflections from reaching the detector. Single wavelength albedo measurements by the IRIS method were performed and averaged over four different positions for each sample. The photovoltaic detector output values could either be read directly by a digital multimeter or stored on a data logger for subsequent retrieval and analysis. A schematic diagram of the experimental set-ups used for spectroradiometric and IRIS measurements is provided in electronic supplementary material, figure S1.

Finally, surface moisture contents of bauxite residues were evaluated using a third optical method, namely, diffuse reflectance infrared Fourier transform spectroscopy (DRIFTS) analyses which were performed with a commercial FTIR spectrometer (Bruker, Vertex 70). The amplitude of the absorption spectral feature centred at 5260 cm$^{-1}$ (i.e. $\lambda \approx 1900$ nm), associated with the $v_2+v_1/v_3$ vibrations (HOH bending + OH stretching combination band) of condensed water in the reflectance spectra of wet bauxite residues, allowing the time-dependent water content at the sample's surface to be monitored [49].

# 3. Results and discussion

## 3.1. Spectral albedo of wet bauxite residues and IRIS measurements

Figure 1*a* and *b* reports representative spectra illustrating the dependence of spectral albedo on the moisture content for wet bauxite residues samples prepared with various solid fractions, as well as for evaporating samples, respectively.

Wet bauxite residues display qualitatively similar absorption features at SWIR wavelengths as other wet granular materials, such as the strong bands that arise from intramolecular vibrational excitations of liquid water centred near 1450 nm and 1900 nm, but they also have the additional particularity of absorbing strongly at 900 nm and below 600 nm due to their elevated iron oxide content [50]. A strong variation in the amplitude of the two strong water absorption spectral features centred at 1450 nm and 1900 nm as a function of solid fraction (figure 1*a*), or time (figure 1*b*), can be noticed. However, spectrophotometric measurements of samples moisture content face significant challenges that arise from the fact that surface TMC may differ from the bulk, as evaporation (and precipitations or sedimentation) can cause mine tailings surfaces to be dryer (wetter) than the bulk [51,52], and from the fact that the spectral albedo depends non-monotonically on TMC.

Indeed, as can be observed in figure 1, the overall reflectance from wet bauxite residues decreases up to solid fractions of 75% ($m_b/m_t$) in prepared samples (figure 1*a*), and up to $t = 30$ min in evaporating samples (figure 1*b*), before increasing again at higher solid fractions, or at later times throughout the evaporation process. This conspicuous, non-monotonic variation in the total reflectance of wet bauxite residues with their moisture content can be attributed to variations in the water layer thickness and surface coverage as well as non-monotonic changes in the scattering of SWIR radiation from the wet bauxite residues grains surfaces. Indeed, changing water layer thickness may induce multiple

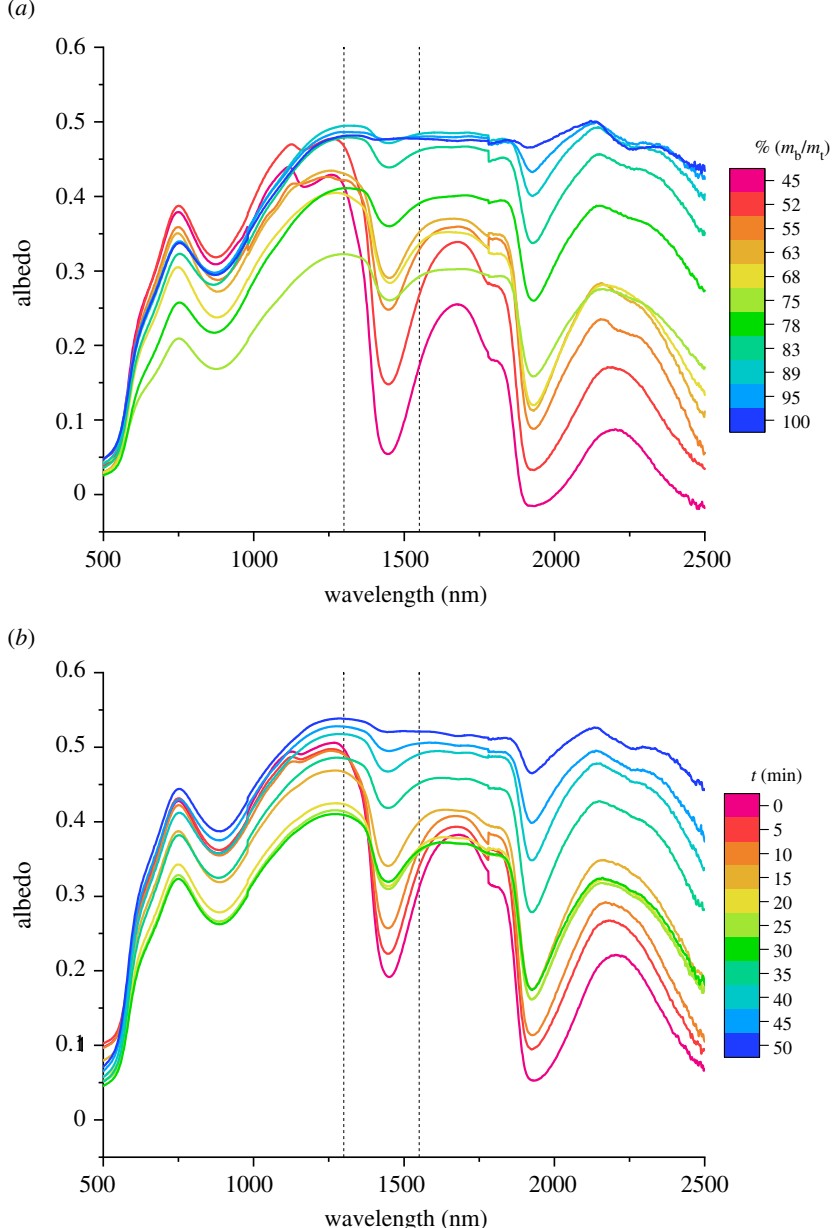

**Figure 1.** Selection of representative spectral albedo data for prepared bauxite residues samples with solid fractions between 45% and 100% ($m_b/m_t$) (*a*). Time series showing the evolution of spectral albedo data for an evaporating bauxite residues sample with an initial solid fraction of 50% ($m_b/m_t$) (*b*). The wavelengths used for the measurements with the IRIS method, that is 1350 nm and 1500 nm, are depicted with the vertical dashed lines.

reflections and refraction, between the rough air/water and water/bauxite residues interfaces, which can result in constructive or destructive optical interferences, along with reflection and scattering of the incident beam [28,35]. Both the evaporating sample and prepared samples show similar modulations in their spectral albedo when their solid fraction approaches 75 % ($m_b/m_t$). This suggests that these effects are rooted in the similar evolutions in the wet bauxite residues surface morphology and topology, as well as water layer thickness, irrespective of whether samples were prepared at a specific solid fraction, or whether they reached this same solid fraction upon evaporation of a sample with an initially greater moisture content.

For the intended application, that is continuous monitoring of surface TMC at mine tailings storage facilities, a simpler proxy than hyperspectral albedo measurements for bauxite residues surfaces drying states and rates would be preferable. For example, one that could be derived directly from either multispectral remote sensing or from single wavelengths ground based measurements and further, that would not suffer from fluctuations in solar irradiance, would be most desirable. As a potentially

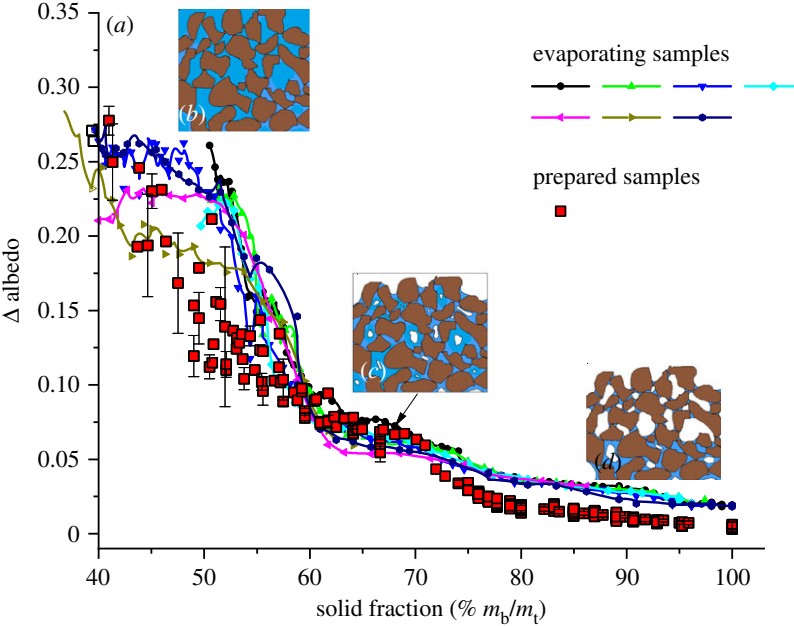

**Figure 2.** Dependence of Δ albedo, calculated from spectral albedo data similar to those presented in figure 1, on sample solid fraction, for bauxite residues samples prepared with different moisture contents (red squares), and for evaporating bauxite residues samples (coloured lines with symbols). (b) Pictograms illustrate the supernatant, (c) capillary and (d) vapour diffusion evaporation regimes.

useful spectroscopic tool of water content at mine tailing surfaces, we chose to probe bauxite residues albedo at 1300 nm (i.e. wavelength at which water absorption is minimal) and at 1550 nm (i.e. wavelength at which water absorption is significant) due to the convenience and availability of laser sources at these telecommunications wavelengths [53]. Δ albedo values, defined as the difference in sample albedo at 1300 nm and 1550 nm, was thus calculated from spectral albedo data, such as those presented in figure 1, and are reported as a function of prepared and evaporating samples solid fractions in figure 2. Note that the uncertainty in Δ albedo for prepared samples is defined by the standard deviation calculated from spectral albedo measurements from eight different samples. Fortunately, although 1550 nm does not correspond to the wavelength of the maximum in the absorption feature from liquid water in this spectral range (i.e. which peaks at 1450 nm; figure 1), figure 2 shows Δ albedo measurements may confer sufficient sensitivity to evaluate wet bauxite residues samples surface moisture content, in addition to convenience and ease of use. Nevertheless, other desirable attributes such as reliability, precision and accuracy must be established for the monitoring of mine tailings surface drying states and rates in the field using the IRIS method (to be discussed in detail further below).

First of all, it can be seen in figure 2 that the Δ albedo values do not display the conspicuous non-monotonic dependence, nor do they reach a minimum value at solid fractions of about 75% neither for prepared samples, nor after 30 min (at which point the initially 50% moisture content sample's solid fraction reaches about 75%) for evaporating samples. Second, as sample surfaces may have a different TMC as the bulk, one must exert caution when trying to estimate samples' moisture content from spectral albedo measurements. Indeed, precipitations and sedimentation may cause the surface moisture content to exceed that of the bulk, yielding an apparent increase in Δ albedo. By contrast, bauxite residues evaporation, arising from low ambient or meteorological relative humidity conditions, could result in a surface layer that is dryer than the bulk yielding an apparent decrease in Δ albedo resulting in an underestimation of the TMC. Therefore, both these phenomena can significantly impact the relationship between the samples Δ albedo and their moisture content. Fortunately, insight into these effects can be gained under the controlled conditions afforded by laboratory investigations.

Indeed, the occurrence of these phenomena can be revealed by comparing Δ albedo values for samples prepared to have a specific solid fraction (red squares) with those from seven samples prepared with different initial solid fractions but that, from then on, were monitored continuously during the dynamic evaporation process (coloured lines with symbols). This comparison reveals that instantaneous Δ albedo and/or solid fractions values for evaporating samples (continuous lines with

symbols) are significantly overestimated compared to those displayed by prepared samples (red squares) for solid fractions smaller than 60%, as well as for solid fractions greater than 70%.

In contrast with the behaviour displayed by prepared samples (figure 2; red squares), the Δ albedo for evaporating samples (figure 2; lines with symbols) remains relatively independent of sample solid fraction until this later reaches values greater than about 50%, whereby its magnitude begins to decrease precipitously with increasing solid fraction up to about 60%. Visual inspection of the samples revealed that a layer of supernatant water formed spontaneously on the surface of residues prepared with solid fractions below 50% due to the rapid sedimentation of the bauxite residues solid fractions as they are left to evaporate (figure 2b). The weak dependence of Δ albedo on solid fraction for evaporating samples prepared with an initial solid fraction lower than 50% must thus arise from the fact that albedo measurements are rather surface sensitive and selective. This causes the reflectivity properties of wet bauxite residues to remain constant (i.e. Δ albedo hovering in the vicinities of 0.25) until a bulk solid fraction of 50% is reached, whereby the supernatant layer has fully evaporated, at which point their Δ albedo decreases precipitously with decreasing moisture content (i.e. increasing sample solid fraction). In prepared samples, for which spectral albedo measurements were acquired immediately after sample preparation, Δ albedo varied continuously and monotonically, from 40% on to higher solid fractions, since samples remained homogeneous during the measurements, as they did not have time to settle in due to sedimentation during the 1 min time delay required to acquire an albedo spectrum.

As solid fractions increase from 50% to 60%, Δ albedo is observed to decrease significantly both for evaporating and prepared samples. Indeed, at these solid fractions, the particles that compose bauxite residues are individually covered with a continuous layer of water (figure 2c) that is responsible for multiple reflections at the air/water and water/bauxite residues interfaces, scattering, as well as absorption by vibrational excitations of water molecules [28,35]. These complex optical effects all contribute to strong modulations in the reflectance of the system with increasing solid fraction, especially so for wavelengths where water absorbs significantly. Therefore, the strong dependence of these optical effects with decreasing moisture content in this range results in a stronger decrease in Δ albedo values with increasing solid fractions for the evaporating bauxite residue samples compared to the prepared samples [54]. This suggests surface moisture content for evaporating samples remains greater than the bulk despite the supernatant layer being essentially absent from the sample surface. Clearly, the bulk moisture content calculated from weighing the samples can significantly underestimate surface moisture content when bauxite residues with solid fractions lower than 60% have time to settle and to suffer sedimentation.

At solid fractions in the 60–70% range, a sudden decrease in the rate of change of Δ albedo with increasing solid fraction is observed, both for prepared and evaporating samples. Furthermore, their Δ albedo values are quite similar suggesting the samples surface and bulk moisture contents must be similar in this range of solid fractions. Previous studies revealed that hydraulic continuity, and therefore, capillary transport in the water film, become increasingly disrupted as samples moisture contents continuously decrease, eventually leaving parts of the bauxite residue's surface bare when the samples solid fractions increase further above approximately 60% [55–58]. Therefore, for similar increments in the samples solid fraction, a smaller variation of absorption by water is observed at the surface (figure 2d) thus leading to a smaller rate of decrease of Δ albedo with increasing solid fraction as a result of incomplete water coverage onto the bauxite residue particles surface.

The differences in Δ albedo values obtained from samples prepared with solid fractions in excess of 70%, and those obtained from evaporating samples that reached the same instantaneous solid fractions, can be explained by the different sample preparation methodology that may have led to a systematic underestimation of the solid fractions for prepared samples. Indeed, samples in the range of 67–100% solid fractions were prepared by adding a given quantity of water to dry bauxite residues to obtain the desired solid fraction and thereafter mixed until the sample was homogeneous. However, samples solid fractions in this range were calculated by neglecting evaporation that may have occurred between each cumulative addition of water, and subsequent measurement, thereby leading to an underestimation of the solid fraction for the prepared sample. Furthermore, evaporation should have a greater effect on sample surface moisture content in the range of solid fraction where hydraulic continuity is disrupted thereby reducing the effectiveness of capillary transport of liquid water to the sample surface. Previous measurements of the evaporative fluxes from wet bauxite residues have been found to vary between 0.2 and 0.6 g m$^{-2}$s$^{-1}$ from quantitative evaluation of evaporation rates as function of sample thickness at an ambient air relative humidity of about 40% [27]. Given the surface area of the Petri dish of $7.85 \times 10^{-3}$ m$^2$, the mass of water evaporated per minute should range from 0.1 g to values as high as approximately 0.28 g. Therefore, this supports the hypothesis that the solid

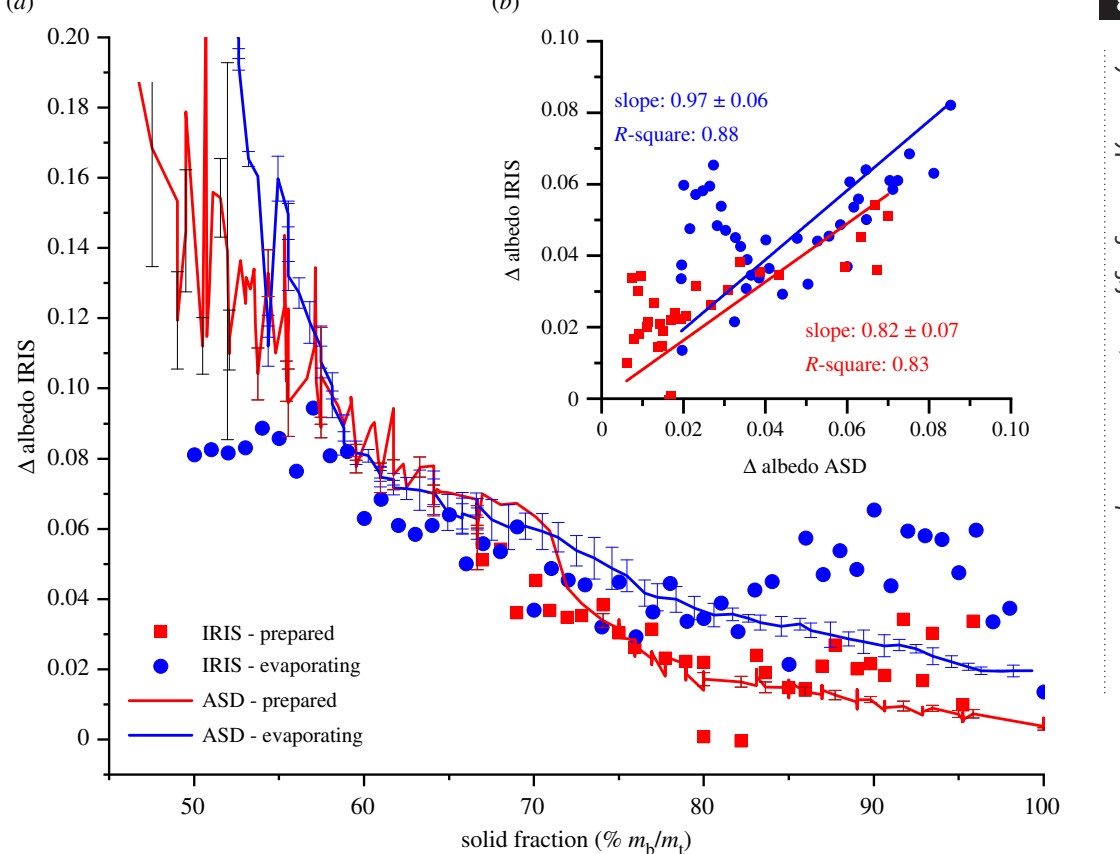

**Figure 3.** (*a*) Dependence of Δ albedo on sample solid fractions, for bauxite residues samples prepared with different moisture contents (ASD: red line; IRIS: red squares), and for evaporating bauxite residue samples (ASD: blue line; IRIS: blue circles). (*b*) Correlation between Δ albedo obtained with IRIS and ASD methods for prepared and evaporating samples.

fractions reported in this range might be significantly underestimated, thereby explaining the lower Δ albedo reported for prepared samples compared to those observed for evaporating samples having solid fractions greater than 70%.

In an effort to evaluate the potential of the IRIS method as a continuous monitoring tool for the surface moisture content in mine tailings, Δ albedo measurements performed at SWIR wavelengths using laser diode sources at 1300 nm and 1550 nm are benchmarked (IRIS: filled symbols), in figure 3*a*, against those acquired using the field spectrophotometer (ASD: continuous lines with error bars), as a function of samples solid fractions, for prepared (red: symbols and lines) as well as for evaporating (blue: symbols and lines) samples. In figure 3*b*, the Δ albedo measured using the IRIS method is correlated with that calculated using spectral albedo measurements obtained with the field spectrophotometer (ASD) at the same instantaneous TMC for the evaporating (blue symbols and linear regression) and prepared (red symbols and linear regression) samples. Linear least square regressions (with the ordinate fixed at the origin) yield coefficients of determination (slopes) of 0.88 (0.97 ± 0.06) and 0.83 (0.82 ± 0.07) for evaporating and prepared samples, respectively.

While the Δ albedo calculated from the prepared samples spectral albedo data (displayed as red squares in figure 2) are simply reproduced here as a red line in figure 3 for convenience and comparison purposes, those from evaporating samples (reported by the seven continuous lines with symbols in figure 2) were averaged for solid fractions greater than 50% and displayed as a blue line in figure 3. Averaging the evaporating samples Δ albedo data from figure 2 affords a greater signal-to-noise ratio thereby allowing the differences in Δ albedo between the prepared (ASD: red line) and the evaporating (ASD: blue line) samples to be better compared in figure 3*a*. Indeed, this enables the significantly diminished Δ albedo values (i.e. by approx. 0.02) displayed by prepared samples compared to evaporating samples with solid fractions greater than 70% to be highlighted. Despite the lower signal-to-noise ratio afforded by the IRIS method (in parts due to variations in measurement-to-measurement sample positioning), this difference is also observed in the single wavelength albedo data acquired from evaporating samples (blue symbols) which display Δ albedo values that are

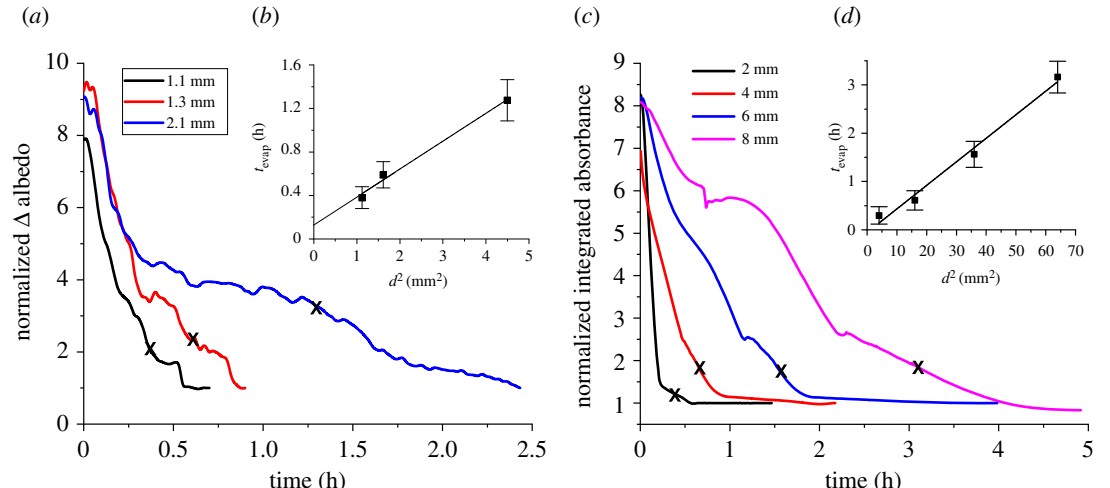

**Figure 4.** Dependence of the normalized $\Delta$ albedo (*a*) and of the normalized integrated absorbance (*c*) with time for evaporating bauxite residue samples that were prepared to have an initial solid fraction of 50%, but that were subsequently monitored continuously as their evaporation proceeded. For ASD measurements (*a*), wet bauxite residues samples thicknesses were 1.1 mm (black line), 1.3 mm (red line) and 2.1 mm (blue line) while for DRIFTS measurements (*c*), sample thicknesses were 2 mm (black line), 4 mm (red line), 6 mm (blue line) and 8 mm (magenta line). Stage II evaporation half-time ($t_{evap}$) as a function of the sample thickness squared ($d^2$) for spectral measurements acquired with the spectroradiometer (*b*) and DRIFT (*d*) suggest late stage evaporation is limited by vapour diffusion.

significantly and systematically enhanced compared to those displayed by prepared samples (red symbols) with solid fractions greater than 70%.

As one can note from figure 3*a*, the $\Delta$ albedo values calculated from measurements obtained with the IRIS method (symbols) show very good correlation with those acquired using the field spectroradiometer (continuous lines with error bars) for solid fractions lower than 80% ($m_b/m_t$) both for prepared (red symbols and line) and evaporating (blue symbols and line) samples. Unfortunately, $\Delta$ albedo values from IRIS measurements become more scattered as sample bulk solid fractions reach over 80% ($m_b/m_t$), especially for evaporating samples (blue symbols). This results in a very significant scatter at $\Delta$ albedo value smaller than 0.04 in figure 3*b*, especially in the evaporating samples data, a phenomenon that can be attributed to the high surface sensitivity afforded by the IRIS method, to the small size of the laser sampling region and, to the very heterogeneous nature of the wet bauxite residues samples surface. Indeed, as the water coverage at their surfaces becomes very uneven at high solid fractions, the focal point of the IR laser beam in the IRIS measurements can sample dry patches or wetter areas where water has remained condensed within bauxite residues internal pore structure, leading to strong variations in the albedo measured at different sample positions by the IRIS method. Therefore, even resorting to artificial light sources in an attempt to improve the reliability, robustness and alleviate unavoidable fluctuations in solar irradiance, the reproducibility of IRIS measurements appears nonetheless compromised by heterogeneities in sample morphology, especially since both prepared and evaporating samples must be displaced back and forth between the albedo and the mass measurements required to monitor their surface and bulk TMC.

## 3.2. Evaporation kinetics

Given the strong correlation between $\Delta$ albedo and the samples solid fractions displayed in figure 3*a*, we are poised to examine how the drying kinetics of wet bauxite residues could be quantified using validation from another spectrophotometric method, namely DRIFTS. $\Delta$ albedo measurements are particularly attractive since, if proven to be robust and reliable, as well as precise and accurate, they could enable validation of remote sensing multispectral and hyperspectral data with continuous monitoring of the drying states and rates of mine tailings surfaces in the field. Therefore, in order to further establish the validity and applicability of the proposed optical sensing tools to assess wet bauxite residues surfaces instantaneous moisture contents, as well as to quantify their drying kinetics, figure 4 compares the time evolution of the normalized $\Delta$ albedo (figure 4*a*) with the time-dependent normalized integrated absorbance of the water IR band at 5260 cm$^{-1}$ obtained from DRIFTS data (figure 4*c*). The normalized $\Delta$ albedo is defined as the ratio of the instantaneous $\Delta$ albedo at time $t$ to

the asymptotic value of the Δ albedo for 'equilibrated' bauxite residues, both obtained with the field spectroradiometer. The resulting kinetic traces are reported in figure 4a for three different sample thicknesses. The normalized integrated absorbance (NIA) is defined as the ratio of the instantaneous integrated absorbance of the water intramolecular $v_2+v_1/v_3$ (HOH bending and OH stretching) combination band centred at 5260 cm$^{-1}$ from the DRIFT spectrum acquired at time t, to the asymptotic value of its integrated absorbance for 'equilibrated' bauxite residues. The resulting kinetics traces are reported in figure 4c for four different sample thicknesses. The time evolution of these simple spectroradiometric indicators, potential proxy for surface moisture contents in mine tailings, reflects the complex interplay between the contributions from the various water transport mechanisms that control the kinetics of the drying process in porous materials such as mine tailings.

As previously reported in the literature [15,27,59,60], water evaporation from porous media can be limited by various water transport mechanisms depending on the samples drying state. Indeed, at the higher moisture contents, supernatant water evaporates freely from the residues surface leading to an elevated drying rate which is relatively independent of solid fraction or time. As seen in figure 4a, once the sample moisture content has decreased until bauxite residues reached solid fractions greater than 60%, evaporation rates become increasingly limited by liquid water transport by capillarity through the pores of the sample [58]. Following the rapid initial evaporation regime, the normalized Δ albedo (figure 4a) and NIA (figure 4c) decrease rapidly with time indicating a transition from free evaporation of the supernatant layer to capillarity-driven transport of water. This can be observed in the first few tens of minutes until the samples normalized Δ albedo reach values of approximately 4 (figure 4a) and until the samples NIA reach values of approximately 2.5 (figure 4c). A second transition in evaporation regime is then later observed and can be attributed to the capillary transport mechanism becoming increasingly disrupted due to the hydraulic continuity being compromised and the percolation threshold being reached at lower water content (less than $32 \pm 4\%$ ($m_w/m_t$), that is as solid fractions greater than 68%) [55,56,58,61]. As lower moisture contents are eventually reached, evaporation becomes increasingly limited by gas diffusion of water vapour through the bauxite residues interconnected pore structure. Vapour diffusion is further hampered by strong water adsorption onto the bauxite residues relatively large specific surface area, along with capillary condensation within the sample's concave pores which increases the tortuosity of the interconnected pore structure [55,62–64]. This causes the effective diffusion coefficients to dwindle and therefore, the evaporation rates to slow down significantly [15,36,57,58,63,65–67].

An estimate of the effective vapour diffusion coefficient, describing water transport through the bauxite residues interconnected pore structure, can be obtained by correlating the evaporation half-times from stage II evaporation ($t_{evap}$, indicated by X on the kinetics traces of figure 4a,c; see definition of $t_{evap}$ in Maurais et al. [27]) with the sample thickness squared, d$^2$ [55,59]. This straightforward analysis follows a protocol described previously for the analysis of drying kinetics using a novel thermal imaging methodology [27] and reproduced here in the insets to the normalized Δ albedo (figure 4b) and NIA (figure 4d) kinetics traces plots. The linear relationship between $t_{evap}$ and $d^2$ strongly suggests drying rates become increasingly limited by slow vapour diffusion in the later stages of the evaporation process [9,15,58,67,68]. The effective diffusion coefficient obtained from the normalized Δ albedo kinetics traces ($D_{eff} = (0.5 \pm 0.4) \times 10^{-5}$ cm$^2$ s$^{-1}$; figure 4b) and from the normalized integrated absorbance kinetics traces ($D_{eff} = (2 \pm 1) \times 10^{-5}$ cm$^2$ s$^{-1}$; figure 4d) are consistent with those obtained previously using a thermal imaging approach [27] showing optical methods, owing to their surface sensitivity and selectivity, are suitable for quantitative measurements of evaporation kinetics from mine tailings surface layers. Furthermore, their high spatial and temporal resolutions are particularly well suited to monitor the time-dependent moisture contents in mine tailings surface layers, both in the laboratory and in the field. These attributes should enable the dependence of evaporation kinetics from mine tailings surfaces on temperature and relative humidity to be quantified, important physico-chemical information to predict the evolution of mine tailings surface moisture contents. Collectively, continuous monitoring of surface drying states and rates, complemented by remote sensing data, should provide crucial parameters helping to forecast the occurrence of fugitive dust emissions from mine tailings storage facilities.

## 4. Conclusion

Optical methods have been demonstrated to possess interesting attributes that could enable remote sensing of surface TMC to be validated by ground base measurements, as they have been shown to display the surface sensitivity and selectivity, precision and accuracy, robustness and reliability,

allowing for continuous instantaneous mining residues surface moisture content to be monitored in real time at mine tailing disposal sites. The methodology described herein, notwithstanding the controlled conditions afforded in the laboratory for prepared and evaporating samples, holds promise to be equally effective under operational and environmental conditions in the field. Instantaneous surface moisture assessment, along with a quantitative analysis of kinetics data, may allow accurate extrapolations of surface drying states and rates, and could thus help predict and prevent the risk of dust emissions. Ongoing investigations at the BRSA should hopefully demonstrate their potential as effective management tools to predict and prevent fugitive dust emissions. Given the availability of low-cost SWIR optical sensors, continuous monitoring of mine tailing storage sites using sensor arrays may contribute to improved remote sensing data yielding accurate risk assessment protocols and improved mitigation strategies for the mining industry.

## Supporting information

A schematic diagram of the experimental set-ups used for spectroradiometric and IRIS measurements are provided.

Data accessibility. The datasets supporting this article are available from the Dryad Digital Repository: https://doi.org/10.5061/dryad.jq2bvq88c [69]. Data also provided in the electronic supplementary material [70].
Authors' contributions. Conceptualization: J.M., F.O., P.A. Data curation: J.M., F.O. Formal analysis: J.M. Funding acquisition: J.M., P.A. Investigation: J.M., F.O. Methodology: J.M., F.O., E.D. Project administration: J.M., P.A. Resources: P.A. Supervision: J.M., P.A. Validation: J.M., F.O., P.A. Visualization: J.M. Writing—original draft: J.M., F.O. Writing—review and editing: J.M., P.A.
Competing interests. We declare we have no competing interests.
Funding. We thank the Vanier Canada Graduate Scholarships, Rio Tinto Alcan, the Fonds de recherche du Québec - Nature et Technologies (FRQNT), the Natural Sciences and Engineering Research Council of Canada (NSERC) Collaborative Research and Development program and the Université de Sherbrooke—Faculté des sciences for grants supporting this work.
Acknowledgement. We are grateful for the technical support and insightful discussions with Rio Tinto scientific staff members, Nicolas-Alexandre Bouchard, Simon Gaboury, Jonathan Bernier and Anderson Santos. Data archiving is underway. We would like to thank Prof. Pedro A. Segura for assistance with statistical analysis of experimental data.

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
