## [Peer Review File · Royal Society Open Science]

Review History

RSOS-210414.R0 (Original submission)

Review form: Reviewer 1

Is the manuscript scientifically sound in its present form?

Yes

Are the interpretations and conclusions justified by the results?

Yes

Is the language acceptable?

Yes

Do you have any ethical concerns with this paper?

No

Have you any concerns about statistical analyses in this paper?

No

Recommendation?

Accept with minor revision (please list in comments)

Comments to the Author(s)

Overall Comment on manuscript # RSOS-210414

The manuscript aimed to use optical method to determine the surface TMC for bauxite residues sampled from BRSA. The authors followed a well structured experimental protocol on selected samples from the preparation toward the analysis using spectroradiometer and sample treatment methods. The results supported the used methods.

minor notes:

31-32 Keywords: it recommended keywords other than these in the article title.

124 it would be great if you draw a schematic diagram of the methodology.

130 and 145 How many samples were prepared and analysed?

198-203 prefer to delete this, it is not a results. It is recommended to start the section with "Figure 1A and 1B ~ continue"

Figure 1: what does the vertical dashed lines represent.

- The results and discussion section was written in a good shape that reflects the methodology.

Best wishes

Review form: Reviewer 2

Is the manuscript scientifically sound in its present form?

Yes

Are the interpretations and conclusions justified by the results?

Yes

Is the language acceptable?

Yes

Do you have any ethical concerns with this paper?

No

Have you any concerns about statistical analyses in this paper?

No

Recommendation?

Accept with minor revision (please list in comments)

Comments to the Author(s)

This is an interesting research. The authors are suggested to revise the manuscript by incorporating the following comments:

1. It would be great to have a map of the bauxite residues storage area (BRSA) used for this research. It would be also useful to have shown the locations of the sampling points on the map.
2. How many total samples were collected from BRSA?
3. How many samples were prepared in the lab?

4. The authors need to explain what was the sampling techniques (random, stratified, grided or clustered) used for collecting the field samples?
5. It would be useful to have some discussions on how the remote sensor arrays should be configured for continuous monitoring of mine tailing storage sites.

Decision letter (RSOS-210414.R0)

Dear Mrs Maurais

On behalf of the Editors, we are pleased to inform you that your Manuscript RSOS-210414 "Monitoring moisture content and evaporation kinetics from mine slurries through albedo measurements to help predict and prevent dust emissions" has been accepted for publication in Royal Society Open Science subject to minor revision in accordance with the referees' reports. Please find the referees' comments along with any feedback from the Editors below my signature. I as Subject Editor support the Associate Editor's recommendation that you consider the comments of the referees carefully and make appropriate minor revision before the paper appears.

Please submit your revised manuscript and required files (see below) no later than 7 days from today's (ie 14-Jun-2021) date. Note: the ScholarOne system will 'lock' if submission of the revision is attempted 7 or more days after the deadline. If you do not think you will be able to meet this deadline please contact the editorial office immediately.

on behalf of Professor Quazi Hassan (Associate Editor) and Peter Haynes (Subject Editor)
openscience@royalsociety.org

Associate Editor Comments to Author (Professor Quazi Hassan):

May I suggest the authors to address the concerns raised by the reviewers.

Reviewer comments to Author:

Reviewer: 1

Comments to the Author(s)

Overall Comment on manuscript # RSOS-210414

The manuscript aimed to use optical method to determine the surface TMC for bauxite residues sampled from BRSA. The authors followed a well structured experimental protocol on selected samples from the preparation toward the analysis using spectroradiometer and sample treatment methods. The results supported the used methods.

minor notes:

31-32 Keywords: it recommended keywords other than these in the article title.

124 it would be great if you draw a schematic diagram of the methodology.

130 and 145 How many samples were prepared and analysed?

198-203 prefer to delete this, it is not a results. It is recommended to start the section with "Figure 1A and 1B ~ continue"

Figure 1: what does the vertical dashed lines represent.

- The results and discussion section was written in a good shape that reflects the methodology.

Best wishes

Reviewer: 2

Comments to the Author(s)

This is an interesting research. The authors are suggested to revise the manuscript by incorporating the following comments:

1. It would be great to have a map of the bauxite residues storage area (BRSA) used for this research. It would be also useful to have shown the locations of the sampling points on the map.
2. How many total samples were collected from BRSA?
3. How many samples were prepared in the lab?
4. The authors need to explain what was the sampling techniques (random, stratified, grided or clustered) used for collecting the field samples?
5. It would be useful to have some discussions on how the remote sensor arrays should be configured for continuous monitoring of mine tailing storage sites.

===PREPARING YOUR MANUSCRIPT===

===PREPARING YOUR REVISION IN SCHOLARONE===

- If you are providing image files for potential cover images, please upload these at this step, and inform the editorial office you have done so. You must hold the copyright to any image provided.
- A copy of your point-by-point response to referees and Editors. This will expedite the preparation of your proof.

- Ensure that your data access statement meets the requirements at <https://royalsociety.org/journals/authors/author-guidelines/#data>. You should ensure that you cite the dataset in your reference list. If you have deposited data etc in the Dryad repository, please only include the 'For publication' link at this stage. You should remove the 'For review' link.
- If you are requesting an article processing charge waiver, you must select the relevant waiver option (if requesting a discretionary waiver, the form should have been uploaded at Step 3 'File upload' above).
- If you have uploaded ESM files, please ensure you follow the guidance at <https://royalsociety.org/journals/authors/author-guidelines/#supplementary-material> to include a suitable title and informative caption. An example of appropriate titling and captioning may be found at https://figshare.com/articles/Table_S2_from_Is_there_a_trade-off_between_peak_performance_and_performance_breadth_across_temperatures_for_aerobic_scope_in_teleost_fishes_/3843624.

Author's Response to Decision Letter for (RSOS-210414.R0)

See Appendix A.

Decision letter (RSOS-210414.R1)

Dear Mrs Maurais,

I am pleased to inform you that your manuscript entitled "Monitoring moisture content and evaporation kinetics from mine slurries through albedo measurements to help predict and prevent dust emissions" is now accepted for publication in Royal Society Open Science.

on behalf of Professor Quazi Hassan (Associate Editor) and Peter Haynes (Subject Editor)
openscience@royalsociety.org

Appendix A

Département de Chimie
Faculté des Sciences
Université de Sherbrooke
Sherbrooke (Québec)

CANADA J1K 2R1

June 21th, 2021.

Prof. Quazi K. Hassan, Subject Editor
Royal Society Open Science
Schulich School of Engineering
University of Calgary
2500 University Dr. NW, Calgary, Alberta T2N 1N4, Canada
E-mail: qhassan@ucalgary.ca

Dear Prof. Quazi K. Hassan

We wish to submit a revised version of manuscript RSOS-210414 entitled:

Monitoring and predicting moisture content and evaporation kinetics from mine slurries through albedo measurements to help predict and prevent fugitive dust emissions

by

Josée Maurais, Frédéric Orban, Emrik Dauphinais, and Patrick Ayotte

for your consideration as a contribution to *Royal Society Open Science*. In this paper, we describe how albedo measurements using short-wave infrared techniques, namely the InfraRed Integrating Sphere (IRIS) method and Diffuse Reflectance Infrared Fourier Transform Spectroscopy (DRIFTS), validated using albedo measurements from a field spectroradiometer, can help monitor the drying state and rates from the surface of mining residues using wet bauxite tailings from a local storage area as a case study. A simple proxy is presented for mine tailings surfaces moisture content and we show how it can provide a robust, precise and convenient tool to monitor drying states and rates from bauxite residues at high spatial and temporal resolution. The methodology described in this manuscript could provide mine tailings storage facilities powerful monitoring and slurry management tools to complement current state-of-the-art remote sensing approaches, supporting the mining industry in their prevention and mitigation efforts by devising improved risk assessment protocols thereby improving prediction and control of fugitive dust scattering events.

Revisions to this manuscript has been detailed below in a point-by-point rebuttal.

With best regards,

Josée Maurais
Département de Chimie, Université de Sherbrooke
Sherbrooke (Québec), CANADA J1K 2R1
Bur.: (819) 821-7889 FAX: (819) 821-8017

Email: Josee.Maurais@USherbrooke.ca

Reviewer #1 (Comments to the Author):

The manuscript aimed to use optical method to determine the surface TMC for bauxite residues sampled from BRSA. The authors followed a well structured experimental protocol on selected samples from the preparation toward the analysis using spectroradiometer and sample treatment methods. The results supported the used methods.

Line 31-32: Keywords: it recommended keywords other than these in the article title

Line 124: It would be great if you draw a schematic diagram of the methodology

Line 130 and 145: How many samples were prepared and analysed?

Line 198-203: Prefer to delete this, it is not a result. It is recommended to start the section with "Figure 1A and 1B... continue"

Figure 1: What does the vertical dashed lines represent

The results and discussion section was written in a good shape that reflects the methodology

Answer to Reviewer #1:

We are thrilled that Reviewer #1 shares our enthusiasm about the work.

Keywords were modified in the manuscript in order to differ from words included in the title:

~~Mine tailings~~ -> Bauxite residues

~~Moisture content~~ -> Tailings moisture content

~~Evaporation kinetics~~ -> Drying kinetics

~~Albedo measurements~~ -> Reflectivity measurements

~~Dust emissions~~ -> Particulate emissions

A schematic diagram of the experimental setup using the spectroradiometer and the IRIS method is now provided in the supporting information (Figure S1).

30 prepared samples were used to acquire the measurements shown in this paper and 8 evaporating samples were prepared and analyzed in this work. The number of samples have been specified at line 163 and 169.

Line 201-204 have been removed and line 204-206 modified to introduce experimental results.

The vertical dashed lines correspond to the wavelengths used with the IRIS method. A short description has been added to the title of Figure 1.

Thank you!

Reviewer #2 (Comments to the Author):

This is an interesting research. The authors are suggested to revise the manuscript by incorporating the following comments:

It would be great to have a map of the bauxite residues storage area (BRSA) used for this research. It would be also useful to have shown the locations of the sampling points on the map.

How many total samples were collected from BRSA?

How many samples were prepared in the lab?

The authors need to explain what was the sampling techniques (random, stratified, grided or clustered) used for collecting the field samples?

It would be useful to have some discussions on how the remote sensor arrays should be configured for continuous monitoring of mine tailing storage sites.

Answer to Reviewer #2:

We are grateful to reviewer #2 for helpful comments and constructive criticism.

Due to confidentiality concerns, we are unfortunately not at liberty to disclose this information.

Over 10 buckets of bauxite residues were collected on the BRSA at different drying states.

30 prepared samples were used to acquire the measurements shown in this paper and 8 evaporating samples were prepared and analyzed in this work. The number of samples have been specified at line 163 and 169.

The sampling technique was random and at different locations on the BRSA. A short description has been added at line 127-128

We are still working on the configuration and communication protocol of the sensors on the storage site. We will surely have a thorough discussion on the remote sensor arrays in our next paper!